# Baloxavir Marboxil: An Original New Drug against Influenza

**DOI:** 10.3390/ph15010028

**Published:** 2021-12-24

**Authors:** François Dufrasne

**Affiliations:** Unité de Microbiologie, Chimie Bioorganique et Macromoléculaire, Département de Recherche et Développement du Médicament, Faculté de Pharmacie, Campus Plaine CP 205/5, Université Libre de Bruxelles, 1050 Brussels, Belgium; francois.dufrasne@ulb.be

**Keywords:** baloxavir marboxil, influenza, baloxavir acid, prodrug, cap-dependent endonuclease, Xofluza

## Abstract

Baloxavir marboxil is a new drug developed in Japan by Shionogi to treat seasonal flu infection. This cap-dependent endonuclease inhibitor is a prodrug that releases the biologically active baloxavir acid. This new medicine has been marketed in Japan, the USA and Europe. It is well tolerated (more than 1% of the patients experienced diarrhea, bronchitis, nausea, nasopharyngitis, and headache), and both influenza A and B viruses are sensitive, although the B strain is more resistant due to variations in the amino acid residues in the binding site. The drug is now in post-marketing pharmacovigilance phase, and its interest will be especially re-evaluated in the future during the annual flu outbreaks. It has been also introduced in a recent clinical trial against COVID-19 with favipiravir.

## 1. Introduction

Among viral diseases, flu is not considered as the most severe. This is the reason why efforts of pharmaceutical research have been focusing on other viruses, especially HIV and hepatitis C, as well as, more recently, on COVID-19 due to the 2020–2021 pandemic. However, in clinical practice, it is well known that flu can be particularly dangerous in disabled, very young, and old people. The development of effective vaccines has increased the somewhat poor interest of the pharmaceutical industries to develop anti-flu drugs. This trend in antiviral research has relatively changed with the observation that the success rate of vaccines is not as high as expected each year, due in part to the high level of mutations of the influenza virus (IV). Finally, it should be also stressed that flu epidemics could be disastrous for human health if it is not tackled adequately. Two facts must be kept in mind: on the one hand, the flu epidemic during the First World War was supposed to have killed more soldiers that the war itself, and, on the other hand, there is the growing fear of a flu pandemic of which the H1N1 outbreak of 2009 was a harbinger, with a high death toll. With this in view, the development of baloxavir is an example of putative new anti-flu drugs that are looming in the future.

Baloxavir (**1**, BX, Figure 1), also called baloxavir acid or BXA, is the active metabolite of the prodrug baloxavir marboxil (BXM, **2**) which is marketed as Xofluza^®^. It is the first approved anti-flu drug since the marketing of the neuraminidase inhibitors (NI) oseltamivir (OMV) and zanamivir in 1999 (USA), laninamivir in 2010 (Japan), and peramivir in 2014 (USA). Older anti-flu drugs include the adamantane derivatives such as amantadine and rimantadine that act on the M2 ion channel and are endowed with more side effects and viral resistance than NIs. Along with side effects, the other growing problem with these compounds is the increase in resistant flu strains that precludes their successful use in clinical practice. The discovery of BX, an inhibitor of cap-dependent endonuclease (CEN) consequently considered as a first-in-class drug, represents a great hope in the fight against seasonal flu. The drug can be used against both influenza A and B viruses (IVA and IVB, respectively).

## 2. Baloxavir Marboxil

### 2.1. Name, Structure, Chemistry, and Rationale Development

The IUPAC name of BXM is ({(12aR)-12-[(11S)-7,8-difluoro-6,11-dihydrodibenzo[b,e]thiepin-11-yl]-6,8-dioxo-3,4,6,8,12,12a-hexahydro-1*H*-[1,4]oxazino[3,4-c]pyrido[2,1-f][1,2,4]triazin-7-yl}oxy)methyl methyl carbonate. The molecule has two stereogenic centers and may exist as four stereoisomers but is marketed as a pure enantiomer (11S, 12aR). To the best of our knowledge, no information is available on the pharmacology of the other stereoisomers.

The molecule was first mentioned in a Japanese patent from Shionogi in 2016 [1]. The prodrug marboxil (also known as S-033188) has been described in subsequent patents from the same company along with other compounds that were overtly designed to circumvent pharmacokinetics (PK) problems. Baloxavir is indeed relatively hydrophilic, and the transport rate across intestinal membrane is low [2]. This hydrophilicity is also related to the acidic character of BX, which originates from a keto–enol structure, leading to a stabilized anion upon deprotonation. 

The conception of BXA stemmed from the work on the rational design of integrase inhibitors (dolutegravir, elvitegravir, raltegravir, and bictegravir), a family of enzyme inhibitors developed against HIV. Flu endonuclease and HIV integrase share the common feature of a bicationic metal ion in their active site (Mn^2+^ for endonuclease and Mg^2+^ for integrase). Owing to the remarkable clinical success of HIV integrase inhibitors, the same strategy has been used to obtain endonuclease inhibitors, with dolutegravir as the model (see Figure 2) [3]. 

The discovery of BXA was the result of a medicinal chemistry campaign organized as follows: screening of metal-chelating compounds using an enzymatic assay on CEN and a cellular phenotypic screen [4]. X-ray diffraction structures of BXA complexed with four different CEN have been obtained: two wild-type enzymes from different flu virus strains (Influenza A/California/04/2009 and Influenza B/Memphis/13/03, PDB codes 6FS6 and 6FS8, respectively) and the same two enzymes but with an I38T mutation (6FS7 and 6FS9, respectively) (see Figure 3). This mutation is considered as the most problematic one, and a rapid method to detect it has been recently developed [5]. Even if the CEN activity of such mutated enzymes is lower, it also renders the protein less sensitive to inhibition by BXA. Analysis of the model from the X-ray structure shows that only Van der Waals interactions occur between BXA and the amino acid residues ALA20, TYR24, LYS34, ALA37, and ILE38. ILE38 is a conserved residue in both IVA and IVB, while the other amino acid residues are different in IVB: THR20, PHE24, MET34, and ASN37 [4]. This is not surprising, since the part of BXA facing the binding pocket has only few polar substituents (F atoms), and amino acids lining the pocket are mainly hydrophobic. Furthermore, this model also explains why I38T mutation renders BXA less active against CEN since the switch between ILE and THR leads to the replacement of a hydrophobic amino acid by a hydrophilic one. The Van der Waals interactions between BXM and this position in the binding pocket are then less favored [4]. These relatively weak interactions are balanced by the high chelating efficiency of the keto–enol moiety toward the Mg^2+^ ions (stabilized on the opposite side by three anionic amino acids—GLU80, ASP108, and GLU119—and the imidazole ring of HIS41). Taking these properties together, BXA is an excellent CEN inhibitor acting at the nanomolar range.

Regarding the production of the molecule, the syntheses of BXM have been extensively reviewed recently by Hughes [6].

### 2.2. Mechanism of Action and In Vitro Characterization

Influenza is a member of the positive sense RNA-virus group and belongs to the *orthomyxoviridae* family. Humans are infected by three viral strains: IVA, IVB, and IVC (IVC is considered as the less dangerous one since it causes only very few infections in humans compared to IVA and IVB). IV infects cells by binding sialic acid residues, which promotes its endocytosis through a cell surface receptor via an enzymatic cascade. After its envelope and the protein involved in the binding to cell membranes are lost, RNA is replicated through a relatively complicated process. Eight segments of viral RNA encode for 11 to 12 proteins only, depending on the strains [7]. Each segment is associated with an RNA-dependent RNA polymerase and other viral nucleoproteins that form a molecular complex called the viral ribonucleoprotein complex [8]. This system works on both replication of viral RNA to produce new virions and transcription to synthesize proteins. RNA-dependent RNA polymerase is formed of different subunits: polymerase acidic protein (PA), polymerase basic protein 1 (PB1), and polymerase basic protein 2 (PB2) [9]. Before functional replicated RNA fragments are obtained, a “cap-snatching” operation must occur. RNA fragments that will serve as primers for the transcription of mRNA are “capped” at their 5′ end by a modified non-coding nucleotide. This capping is essential for the regulation of transcription and the production of mature mRNA expected to remain stable, especially toward RNA degradation enzymes such as ribonucleases. This part of the nucleic acid has to be removed, and this operation is catalyzed by the endonuclease activity of the N-terminal region of PA, namely, the “cap-snatching” process. It starts by the binding of PB2 to the capped region of RNA. Subsequently, the fragments are copied by the PB1 part of the ribonucleoprotein complex, and, finally, the new viral envelope is assembled and the virions are liberated through the action of neuraminidases, a series of enzymes able to cut the remaining bonds between sialic acid and the receptor on the viral envelope.

In view of the biology of IVs, the available putative targets to develop new drugs are relatively few, due to the “simple” mechanisms involved in viruses replication. However, owing to the relatively good molecular characterization of flu endonuclease, this enzyme has been identified as a potential new target for drug development. In the first in vitro enzymatic assays conducted, BXA selectively inhibited CEN without affecting the other component of the enzymatic complex [3]. As expected, the prodrug BXM had no activity. Inhibition potency was confirmed by the low content in the different types of viral RNAs (mRNA, vRNA, and cRNA). One should logically assume that the lack of RNA impedes the production of new virions. However, the mechanism of viral decay is more subtle and is initially caused by inhibition of protein synthesis that, consequently, leads to the impossibility to form envelopes for the new virions [10]. More interestingly BXA is able to inhibit CEN from IVA and IVB with different potencies (activities were 2- to 12-fold smaller against IVB), and also enzymes from other flu strains [11]. This observation is important since it is expected that flu viruses commonly infecting humans can hybridize with other generally harmless virus subtypes [12]. This is due to the highly conserved amino acids sequence amongst the different viral variants. The only limitation in activity was observed with CEN, showing an I38T mutation (see above) that leads to a 30- to 50-fold decrease in CEN inhibition potency with a greater extent in IVA than in IVB infections. This mutation has been observed in rates ranging from 1 to 20% in phase II and pediatric studies, respectively [4]. In the CAPSTONE-1 study (see below), the mutation rates were of 2.2 and 9.7% in phases II and III, respectively. These mutations were related to longer treatment times to achieve curing. More recent analyses confirmed these observations [13].

An in vitro study on a large number of virus variants from Asia, between 2012 and 2018, was concluded in 2018. The aim of the authors was to determine the extent of putative resistance of several viral strains toward BXA [14]. A total of 218 viruses were tested (including IVA and IVB), all of them having been responsible for flu infections during the mentioned period in Asia. In vitro EC_50_ were sixfold higher for IVB than for IVA, which could be explained by differences in the key amino acid residues (see Section 2.1). This trend was also observed with OMV. The other important result was the larger range of EC_50′_s for IVB than for IVA, showing a wider variability in the chemical environment in the binding site.

### 2.3. Preclicinal and Clinical Trials, Toxicology, and Pharmacokinetics

The first studies on animal models included flu-infected mouse, in which rapid reduction of the blood viral count and mortality were noted [15]. A more recent preclinical trial was published at the end of 2018. This study was undertaken on mouse to assess the potential synergistic effect of BXM with OMV on a model of IVA [16]. First, BXM was used alone, and an analysis of the inflammatory markers of the lungs showed lower levels of cytokines (IL-6, MCP-1, MIP-1α, and INF-γ) and inflammatory cells (macrophages and neutrophils). This has not been observed with OMV alone. This effect is worth noting since, according to the authors, it could be linked to a better preservation of the physiological structure of the lungs. Concerning the association of BXM and OMV, the survival rates of infected mouse were better for the combination of OMV with suboptimal doses of BXM than with OMV alone, and positive effects on body weight were also noted. However, the best results were obtained with optimal doses of BXM.

Initial phase I clinical trials have been conducted on male Japanese volunteers with BXM [17]. Two groups were formed: the first one to study dose escalation from 6 to 80 mg versus placebo, the other one to investigate the influence of food on the PK. BXM was rapidly hydrolyzed into BXA most probably by plasma esterases, but also in both intestinal lumen and epithelium, and also in liver [6]. The authors estimated from nonclinical studies that a plasma concentration dosing after 24 h (C_24_) of 6.85 ng/mL would be enough to exert antiviral effect. This level was reached with the smallest doses used (6 mg), giving rise to a C_24_ of 6.93 ng/mL. Fasted people had slightly higher concentrations than fed ones, showing a significant effect of food on the resorption of BXM (C_max_ decrease of 48% and AUC decrease of 36%) [18]. However, the package insert mentioned no recommendations for taking the drug with or without food, and this was in line with the conclusion of a study on PK showing discrepancies between results obtained in phases I, II, and III regarding the effect of food on BXM’s PK [19]. There was a linear correlation between the dose and the C_24_, a characteristic feature of simple diffusion transport (non-saturable). After metabolization of BXM, BXA has a long half-life (t_1/2_ = 90.9 h for a dose of 6 mg). Another effect of food could be caused by its content in metallic ions. As it is the case for other drugs with similar structures (for instance quinolones antibiotics and HIV integrase inhibitors, the latter having served as starting molecules for the design of BXA), the keto-enol group in BXA can form chelates with multivalent ions such as Ca^2+^, Mg^2+^, Fe^3+^, and Zn^2+^, for example. Resulting complexes are much less absorbed in the intestine, leading to a decreased bioavailability. However, this effect is logically expected to be lower for BXM than for the drugs cited above since BXM is a prodrug that does not possess the keto–enol moiety, and only hydrolyzed BXM in the intestinal lumen could be subjected to this problem. Nevertheless, practitioners should remain cautious since, to our knowledge, the relative hydrolysis rates in intestine and blood are not available, and the precise extent of BXM hydrolysis in the intestinal tract should not be underestimated. Finally, there were no significant differences between genders in patients with impaired liver function [19]. 

Regarding side effects, the molecule was well tolerated [17]. Only the following observations were made in both groups, and their frequency did not increase with dose escalation: headache (most frequent) and perturbations of blood composition (increase in alanine and aspartate aminotransferase, lactate dehydrogenase, white blood cells, and eosinophil counts). All these events were reversible after trial completion. It was concluded that the tolerance profile was excellent, with no severe adverse events or treatment-related deaths reported. Clinical data over side effects, however, showed rather questionable results: only volunteers receiving the lowest doses (6 mg) reported these effects, while at higher doses (up to 80 mg), no such observations were made. The authors provided no tentative explanations for this. Furthermore, the subsequent CAPSTONE-1 study (phases II and III) provided similar results [15]. To be complete, it should be mentioned that in the package insert and other references based on communications from Shionogi, the most significant side effects were as follows: diarrhea (3%), bronchitis (2%), nausea (1%), nasopharyngitis (1%), and headache (1%), all of them in the same range as for the placebo. Interestingly, the incidence of these effects was lower than in patients treated with OMV.

The PK of BXM has been studied (data from Shionogi) [10,11]. As expected, BXM is hydrolyzed into BXA in the small intestine but also in blood and liver by arylacetamide deacetylase. In the body, BXA is mainly metabolized by uridine monophosphate glucuronosyl transferase 1A3 (UGT1A3) into its glucuronidated form (on the –OH group of the enol). Only a very small amount is oxidized by cytochrome 3A4 (CYP) (S to sulfoxide), and BXA weakly inhibits this CYP as well as CYP 2B6 and 2C8 [11]. Due to enterohepatic circulation, the glucuronide metabolite is excreted into the gastrointestinal tract where the glucuronic acid is removed by glucuronidases. Consequently, BA is found in feces at 80% of the administered dose (15% in urine), with an estimated mean elimination half-life of 96 h. The drug has linear PK with a long elimination half-time (80–100 h) after a single oral administration. The protein binding is high (93–95%), and the distribution volume extremely large (494–625 L). Finally, it is also worth mentioning that both BXM and BXA are weak inhibitors of P-glycoprotein transporters.

Another important element to investigate was the possibility of using BXM synergistically with other anti-flu drugs. The safety of such association has been first studied in a phase I trial using a combination of BXM and OMV [10]. A non-clinical study showed that this synergy was effective and safe, with neither detrimental drug–drug interactions nor side effects worsening. Interest of using BXM with OMV is that the latter (i) is metabolized by enzymes other than BXM and BXA, (ii) is not a substrate for CYP, and (iii) is excreted in urine. This lack of common metabolic pathways is interesting, with the only concern regarding the fact that OMV is a substrate of P-glycoprotein transporters. Despite the study being relatively poor in terms of number of patients enrolled (18) and investigated parameters, the authors concluded that association was safe and effective to treat seriously ill patients. 

A series of phase II and III trials have been undertaken. In the phase III study (CAPSTONE-1), children and adults with uncomplicated IVA or IVB infections were enrolled. The trial design comprised three groups: one received 40 to 80 mg of BXM, another one OMV 75 mg, and the last one a placebo, all twice daily. The effect of BXM on the duration of flu symptoms was significantly higher for the BXM groups than for the placebo group (53.7 h vs. 80.2 h, respectively). The viral shedding was also considerably reduced with BXM than with OMV and placebo (24 h, 72 h, and 96 h, respectively) [15]. One important conclusion was that starting the treatment early (less than 24 h) after the appearing of initial flu symptoms, as is the case for NIs, gave especially good results. Some differences were also noted in the responses between patients from Japan or the United States regarding the time needed to see symptom alleviation, but these were ascribed to discrepancies in health management in both countries, an explanation that has been criticized recently (see the Conclusion section). Even if no direct comparisons between BXM and NI (namely, OMV) have been done during clinical trials, the authors tried to compare them from previous clinical data. From these observations, it was concluded that BXM is superior to NI in decreasing the blood viral load. Despite this, the time requested for symptom alleviation was similar for groups treated with BXM and NI, even if a synergistic effect was observed (see above). No explanations have been given for this phenomenon. 

In another study, particularly interesting features were observed regarding PK [19]. The effect of body weight and race were investigated, being found to be relevant for the dose calculation regarding differences in clearance and volumes of distribution in the central compartment. Race differences could not be fully explained up to now, although it has been suggested that discrepancies in intestinal absorption and metabolization rates by UGT1A3 could be responsible for this. Finally, the authors concluded that dosing should be calculated on the basis of body weight instead of genetic features of the patient with regard to their race, since the latter have a smaller influence on PK. These recommendations are used today in clinical practice (see Section 2.4). 

Regarding IVA and IVB, despite the significant differences in amino acid residues in the binding site, and besides the study of Hayden et al. [15], results from CAPSTONE-2 trial have shown that the clinical effects were seemingly the same on both strains [20]. Indeed, BXM was superior to placebo on both strains and similar and superior to OMV on IVA and IVB, respectively.

To the best of our knowledge, nine clinical trials involving BXM are mentioned in the clinical trial databases (see Table 1). 

### 2.4. Uses

Baloxavir and its prodrug have a strong antiviral activity in vivo against IVA and IVB, even on NI and M2 channel blocker-resistant strains [11,12]. BXM has been approved as 20 or 40 mg tablets for the treatment of uncomplicated influenza as well as for post-exposure prophylaxis in patients aged 12 years and above [18]. It is manufactured by the Japanese firm Shionogi and distributed by Genentech, a member of the Roche group [11]. Recommended doses are 40 and 80 mg in a single regimen for patients weighing less and more 80 kg, respectively. 

### 2.5. Medical Interest

Due to its relative recent use in medicine, it is still difficult to provide a strong statement on the interest of BXM [21]. The first approval was in Japan (February 2018), followed by the USA (October 2018), then in Europe (January 2021) for “treating and preventing flu in adults and children from 12 years of age”. For instance, owing that European Medicines Agency (EMA) considers thata new active substance should be monitor for 5 years, BXM is still a very young drug on the market. 

Concerning the early data available on the drug, some conclusions found both in clinical trial papers and review articles are not in agreement, especially regarding the efficacy of BXM compared to that of OMV. Advantages over other anti-flu drugs have been noted such as the single dose compared to multiple dose regimen, the good activity on resistant strains, or the low level of side effects compared to OMV. However, resistances could arise in the future if BXM is too extensively prescribed, especially without limiting its use strictly to the most fragile patients.

It is therefore essential to continuously analyze available clinical data on BXM, to the largest extent, regarding its efficacy, safety, and propensity to lead to flu variants. A recent meta-analysis on the clinical efficacy and safety of BXM provides some evidence about its interest [22]. It has a better virological response than OMV and is clinically as effective as the latter against flu, while being safe compared to both OMV and placebo. The remarkable feature of this analysis is that it encompasses a large number of countries from Asia, Europe, and the Americas. Another study, limited to Japanese flu outbreaks in 2019–2020, showed better results for BXM compared to OMV (median duration of fever of 22.3 and 27.5 h, respectively). Appearing of variants was only noticed during the early post-treatment stage [23].

Contrary to scientists and clinicians, health authorities are very cautious about this new drug. Such a position is generally adopted by these institutions for two reasons: (1) their responsibility could be questioned in cases of underestimated problems and (2) countries organizing a health insurance system should carefully analyze the balance between the price and the benefits/risk ratio for the patients. This is the case for flu, which is not a severe disease in most cases but an acute condition that could, however, decrease economic activity because of the absence of ill workers For instance, in Belgium, the Belgian center for pharmacotherapeutic information (BCPI) has stated that the use of BXM (as well as OMV) should be tightly restricted to people with the most serious life-threatening conditions and that, for flu prophylaxis, these drugs cannot replace vaccination. Moreover, the BCPI concluded that there is no scientific evidence that BXM has positive effects on complications such as pneumonia or patient mortality. However, some positive clinical observations have been published since the beginning of 2021. A recent clinical trial in Japan in children between 6 and 12 concluded that BXM was efficacious and safe for Japanese patients with influenza infection [24]. A comparison of the antiviral medicines in Japan showed that BXM reduced the rates of complications in patients suffering from flu [25]. The main concern remains the development of resistance, for which insufficient data are available for the moment. For instance, in Japan, a high frequency of I38T variants with decreased susceptibility to BXM has been reported [26]. Another study showed that resistance is more problematic in patients treated with BXM compared to OMV [27]. The monitoring program of the World Health Organization would follow the spread of resistant flu viruses in the world to assess the extent of the problem.

To conclude this section, we still need more time (probably no less than 2 years) and several flu episodes in different countries to have a more complete view on the potential interest of BXM on human health, as for other innovative drugs. 

## 3. Conclusions

Baloxavir marboxil is clearly an antiviral agent that needs to confirm its interest as treatment against flu, despite the good results obtained during the clinical trial evaluations and its clinical use in Japan; the USA; and, more recently, Europe. Indeed, BXM is still currently in the pharmacovigilance phase (phase IV) in these countries. Recently, a critical analysis of the CAPSTONE-1 trial revealed that the use of enrichment process during the study could have overshadowed the poor effect of BXM in American patients compared to Japanese ones [28]. Analysis of the CAPSTONE-2 study, together with the most recent clinical data on a very large number of patients, is expected to fill this gap [29]. Finally, attention must be kept on the development of flu variants prone to resisting to this new drug.

The future story of BXM is also linked to the present coronavirus (2019-nCoV) worldwide epidemic since it is subjected to a clinical trial in China (clinical trial code at the Chinese Clinical Trial Register: ChiCTR2000029544), together with favipiravir to assess its potency against this life-threatening disease [30].

## Figures and Tables

**Figure 1 pharmaceuticals-15-00028-f001:**
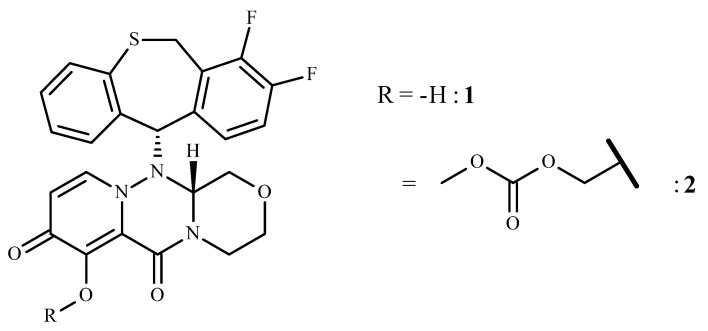
Structures of baloxavir (**1**) and its prodrug baloxavir marboxil (**2**).

**Figure 2 pharmaceuticals-15-00028-f002:**
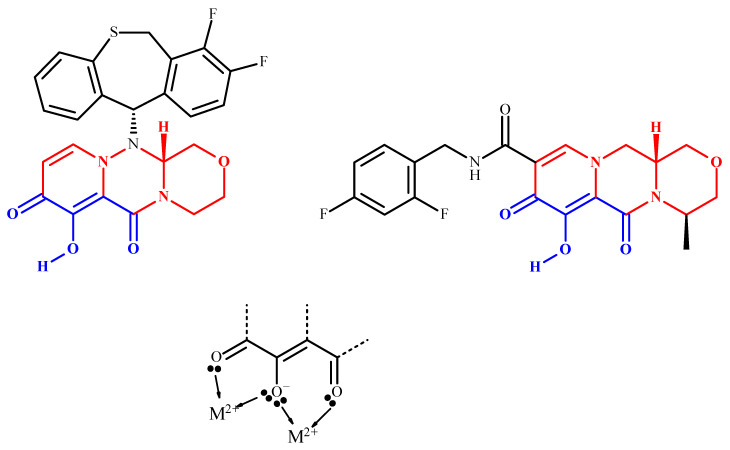
Structures of BXA (**left**) and dolutegravir (**right**), showing the similar tricyclic parts (red) and the metal binding keto–enol moiety (blue). Two bivalent metal ions (either Mn^2+^ or Mg^2+^ in the enzymes, symbolized by M^2+^) can be chelated by the molecular pincers (**below**).

**Figure 3 pharmaceuticals-15-00028-f003:**
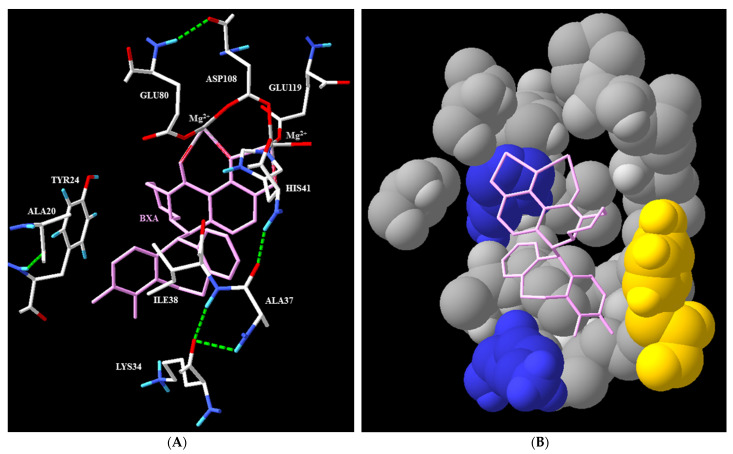
(**A**) Model obtained using SwissPDBViewer ^®^ 4.1.0. from the X-ray diffraction structure analysis of BXA (pink) complexed in Influenza A/California/04/2009 (pH1N1) endonuclease (PDB code: 6FS6) [4]. (**B**) A reverse view of the binding site showing the hydrophobic character of the BXA binding site (color code for amino acids: hydrophobic ones in grey, basic ones in blue, and aromatic ones in yellow).

**Table 1 pharmaceuticals-15-00028-t001:** Ongoing and completed clinical studies on BXM in chronological order ^1^.

Codes and Completion Dates ^2^	Title	Status	Main Clinical Results—Trial Characteristics ^3^
NCT02954354 (4/17) [15]	A study of S-033188 (BXM) compared with placebo or OMV in otherwise healthy patients with influenza (CAPSTONE 1)	Completed	Sponsor: Shionogi (Japan). Countries: not provided. Main outcomes: see [15].
2020-000696-20 (3/19)	A phase 3 randomized, double-blind, placebo-controlled study to confirm the efficacy of a single dose of BXM in the prevention of influenza virus infection	Completed	Sponsor: Shionogi (Japan). Country: Japan. 749 patients reenrolled: 374 received BXM and 375 received placebo. Main outcomes: proportion of subjects who are infected with influenza virus (RT-PCR positive), and present with fever and at least one respiratory symptom.
2018-002169-21 (4/19)	A multicenter, randomized, double-blind, active (OMV)-controlled study to assess the safety, PK, and efficacy of BXM in otherwise healthy pediatric patients 1 to <12 years of age with influenza-like symptoms	Completed	Not available yet. Sponsor: Hoffmann-La Roche (Switzerland). Countries: USA. 173 patients enrolled in two groups: 115 received BXM and 58 received OMV. Main outcomes: percentage of participants with adverse events and serious adverse events and PK data.
NCT03629184 (4/19)	Study to assess the safety, PK, and efficacy of BXM in healthy pediatric participants with influenza-like symptoms	Completed	Sponsor: Hoffmann-La Roche (USA). Countries: Costa Rica, Israel, Mexico, Poland, Russian Federation, Spain, USA. 173 patients enrolled, 115 in the BXM group, 58 in the OMV group. Main outcomes: PK data, time to alleviation of symptoms, time to cessation of viral shedding.
NCT03959332 (7/19)	Study to assess the PK, safety, and tolerability of BXM in healthy Chinese participants (phase I)	Active Not recruiting	Sponsor: Hoffmann-La Roche (China). In collaboration with Shionogi. Country: China. 32 patients enrolled, 16 in two groups (40 and 80 mg). Main outcomes: PK data, adverse effects.
NCT02949011 (11/19) [20]	Study of S-033188 (BXM) compared with placebo or OMV in otherwise healthy patients with influenza (CAPSTONE-2)	Completed	Sponsor: Shionogi. Countries: worldwide. Main outcomes: see [20].
2021-001026-22 (3/20)	An open-label study to assess the safety, tolerability, pharmacokinetics, and efficacy of BXM 2% granules after administration of a single dose to otherwise healthy pediatric patients with influenza	Completed	Sponsor: Shionogi. Countries: Japan. 45 patients enrolled. Divided in two groups: 9 patients (weight < 10 kg) received 2 mg/kg of BXM and 36 (weight ≥ 10 kg and < 20 kg) received 20 mg of BXM. Main outcomes: time to alleviation of influenza illness.
NCT03684044 2018-001416-30 (3/20)	A phase III, randomized, double-blind placebo-controlled, multicenter study to evaluate the efficacy and safety of BXM in combination with standard-of-care NI in hospitalized patients with severe influenza	Completed	Sponsor: Hoffmann-La Roche (USA). Countries: worldwide. 363 patients enrolled, 239 in the BXM group, 124 in the placebo group. Main outcomes: clinical improvement and PK data.
NCT04141917 (3/21)	Test-and-treat for influenza in homeless shelters	Terminated	Reason of the termination was operational futility.
NCT04141930 (4/21)	Pilot of cohort of households for influenza monitoring and evaluation in Seattle (pCHIMES)	Completed	Sponsor: University of Washington (USA). In collaboration with Genentech (USA). Country: USA. 481 patients enrolled in two groups: 302 patients were eligible to receive either 40 or 80 mg single dose oral BXM and 179 were not eligible to receive on-label use of BXM due to age or medical history. Main outcome: symptom onset after drug administration up to 48 h.
NCT03653364 (8/22)	Study to assess the safety, PK, and efficacy of BXM in healthy pediatric participants from birth to < 1 year with influenza-like symptoms	Recruiting	Sponsor: Hoffmann-La Roche (USA). Countries: worldwide. Main outcomes: PK data, time to alleviation of symptoms, time to cessation of viral shedding.
NCT04327791 (12/22)	Combination therapy with baloxavir and oseltamivir 1 for hospitalized patients with influenza (The COMBO Trial 1) (COMBO 1)	Recruiting	Sponsor: Bassett Healthcare (USA). In collaboration with: Genentech (USA) and Viroclinics Biosciences (the Netherlands). Phases II and III. Main outcomes: time to clearance of viral shedding.
NCT03969212 (2/23)	Study to assess the efficacy of BXM versus placebo to reduce onward transmission of influenza A or B in households	Recruiting	Sponsor: Hoffmann-La Roche (USA). Countries: worldwide. Main outcomes: virological transmission by day 5.
NCT04712539 (3/23)	BX and OMV for the treatment of severe influenza infection in immunocompromised patients	Not yet recruiting	Sponsor: M.D. Anderson Cancer Center (USA). Country: USA. Main outcomes: changes in viral loads, incidence of complicated hospital stay.
NCT05012189 (12/23)	BX versus OMV for nursing home influenza outbreaks	Recruiting	Sponsor: Insight Therapeutics (USA). Country: USA. Main outcomes: To demonstrate the non-inferiority of prophylactic BX vs. OMV to prevent influenza-like illness (ILI) after an index influenza case in nursing homes.

^1^ Data from ClinicalTrials.gov and EU Clinical Trials Register. ^2^ Actual or estimated (month/year). ^3^ When more than 10 countries are involved, the name of a continent or the term “worldwide” are empirically used.

## Data Availability

Data sharing not applicable.

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
