# Peer review of "Baloxavir Marboxil: An Original New Drug against Influenza"

_pharmaceuticals, 2021, doi:10.3390/ph15010028_

Round 1
Reviewer 1 Report
The review article is describing the chemistry, mechanism of action and clinical trials of Baloxavir marboxil, a new antiviral drug against seasonal flu
infection. The author gave details over different aspects related to drug structure, action, and uses.
1- looking in pubmed, one can see that there is at least another review article having the same name. I strongly recommend to modify the title of this review article.
2- English needs to be revised at different places.
3- Line 128: usually the word "metabolism" is not used with viruses; instead the word "replication".
Author Response
I thank reviewer 1 for taking the time to review this manuscript.
Here are the responses to his/her comments:
1- looking in pubmed, one can see that there is at least another review article having the same name. I strongly recommend to modify the title of this review article.
The title of the manuscript has been changed (“Baloxavir marboxil : an original new drug against influenza “) so that there will be no confusion with previous papers.
2- English needs to be revised at different places.
The language will be improved throughout the manuscript with the help of the editorial office.
3- Line 128: usually the word "metabolism" is not used with viruses; instead the word "replication".
I thank reviewer 1 for the relevance of this remark. The correction has been made.
Yours faithfully.
FRANCOIS DUFRASNE

Reviewer 2 Report
Thank you for allowing me the opportunity to review your contribution to Pharmaceuticals. The article is written in a very concise form and well-organized fashion such that those who are not deeply involved in medicinal chemistry would be able to understand the story of BM's development. As such, this reviewer has very little to critique on the current version.
While the overall manuscript would be improved by data/statements on its impact in comparison to current treatments, this reviewer understands that such data does not yet exist...which is slightly surprising considering the approval in Japan in 2018. This could be a minor addition under Section 2.5 to denote any medical rationale for lack of use (and knowing the data is going to predominantly come from Japanese populations).
Author Response
I thank reviewer 2 for taking the time to review this manuscript.
Here are the responses to his/her comments:
While the overall manuscript would be improved by data/statements on its impact in comparison to current treatments, this reviewer understands that such data does not yet exist...which is slightly surprising considering the approval in Japan in 2018. This could be a minor addition under Section 2.5 to denote any medical rationale for lack of use (and knowing the data is going to predominantly come from Japanese populations).
This remark is strongly relevant and I strived to improve this part of the manuscript. I reorganized and updated section 2.5 in order to highlight the evolution of knowledge on the use of baloxavir marboxil in clinics. The section comprised the following paragraphs respectively with new references :
- The “history” of baloxavir approval.
- The very first clinical data on the drug.
- The most recent data available on the clinical efficacy, the comparison with oseltamivir and the appearing of variants.
Kuo, Y.-C.; Lai, C.-C.; Wang, Y.-H.; Chen, C.-H.; Wang, C.-Y. Clinical efficacy and safety of baloxavir marboxil in the treatment of influenza: A systematic review and meta-analysis of randomized controlled trials. J. Microbiol. Immunol. Infect., 2021, 54, 865-875.
Ishiguro, N.; Morioka, I.; Nakano, T.; Furukawa, M.; Tanaka, S.; Kinoshita, M.; Manabe, A. Clinical and virological outcomes with baloxavir compared with oseltamivir in pediatric patients aged 6 to < 12 years with influenza: an open-label, randomized, active-controlled trial protocol. BMC Infect. Dis., 2021, 21, 777.
- The position of health authorities showing both the cautious attitude of medicine agencies and the new positive clinical results gathered during the post-approval period, but also the concerns about the most problematic variants (I38T).
Chong, Y.; Kawai, N.; Tani, N.; Bando, T.; Takasaki, Y.; Shindo, S.; Ikematsu, H. Virological and clinical outcomes in outpatients treated with baloxavir or oseltamivir: A Japanese multicenter study in the 2019-2020 influenza season. Antivir. Res., 2021, 192, 105092.
Liu, J.-W.; Lin, S.-H.; Wang, L.-C.; Chiu, H.-Y.; Lee, J.-A. Comparison of Antiviral Agents for Seasonal Influenza Outcomes in Healthy Adults and Children: A Systematic Review and Network Meta-analysis. JAMA Netw. Open, 2021, 4, e2119151.
Takashita, E. Influenza polymerase inhibitors: mechanisms of action and resistance. Cold Spring Harb. Perspect. Med., 2021, 11, a038687.
Sato, M.; Takashita, E.; Katayose, M.; Nemoto, K.; Sakai, N.; Fujisaki, S.; Hashimoto, K.; Hosoya, M. Detection of variants with reduced baloxavir marboxil and oseltamivir susceptibility in children with influenza A during the 2019-2020 influenza season. J. Infect. Dis., 2020, 222, 121-125.
As you mentioned, most of the data are limited to Asia.
I hope that these modifications will meet the requirements of reviewer 2 for improving this part of the manuscript.
Yours faithfully..
FRANCOIS DUFRASNE
